# Sperm Cryopreservation as a Tool for Amphibian Conservation: Production of F2 Generation Offspring from Cryo-Produced F1 Progeny

**DOI:** 10.3390/ani13010053

**Published:** 2022-12-23

**Authors:** Shaina S. Lampert, Isabella J. Burger, Allison R. Julien, Amanda B. Gillis, Andrew J. Kouba, Diane Barber, Carrie K. Kouba

**Affiliations:** 1Department of Biochemistry, Molecular Biology, Entomology, and Plant Pathology, Mississippi State University, Mississippi State, MS 39762, USA; 2Department of Wildlife, Fisheries, and Aquaculture, Mississippi State University, Mississippi State, MS 39762, USA; 3Department of Ectotherms, Fort Worth Zoo, Fort Worth, TX 76110, USA

**Keywords:** anuran, caudate, hormone therapy, in vitro fertilization

## Abstract

**Simple Summary:**

Assisted reproductive technologies are key components in augmenting captive breeding programs for at-risk amphibian species. Although numerous studies show that these reproductive technologies are successful in generating offspring, there is very little research investigating the reproductive viability of those individuals. It is crucial to gain insight on the reproductive potential of offspring generated using assisted reproductive technologies when those individuals are to be used for future breeding, or wild release. We report three instances where amphibian offspring generated using a suite of assisted reproductive technologies were monitored through reproductive maturity and subsequently produced an F2 generation.

**Abstract:**

Sperm cryopreservation and biobanking are emerging as tools for supporting genetic management of small and threatened populations in amphibian conservation programs. However, there is little to no evidence demonstrating reproductive maturity and viability of offspring generated with cryopreserved sperm, potentially limiting widespread integration of these technologies. The purpose of this report is to demonstrate that amphibian sperm can be cryopreserved and thawed to successfully produce individuals of an F1 generation that can reach adulthood and reproductive maturity, to generating viable gametes and an F2 generation. Species-specific exogenous hormones were administered to both F0 and F1 adults to stimulate spermiation and oviposition in the eastern tiger salamander (*Ambystoma tigrinum*), dusky gopher frog (*Lithobates sevosa*), and Puerto Rican crested toad (*Peltophryne lemur*). Sperm cells collected non-lethally from F0 adults were cryopreserved, thawed, and used for in vitro fertilization (IVF) to produce F1 offspring. Individuals of the F1 generation are shown to reach adulthood, express viable gametes, and produce offspring through facilitated breeding, or IVF. The production of amphibian F2 generations shown here demonstrates that amphibian sperm collected non-lethally can be banked and used to generate reproductively viable animals of subsequent generations, thus maintaining valuable genetic linages and diversity in threatened amphibian species. The incredible value that cryopreservation of sperm has for long-term genetic management aids in the sustainability of both in situ and ex situ conservation efforts for this taxon.

## 1. Introduction

Amphibian assisted reproductive technologies (ART) are commonly being used in conservation breeding programs to manage for lack of natural reproduction, low reproductive output, and declining genetic diversity [1,2]. For example, nearly all managed species survival programs in U.S. zoos utilize hormone therapy for induced breeding with either one or both sexes receiving treatment with gonadotropin releasing hormone analog (GnRHa), human chorionic gonadotropin (hCG), or a combination of both to stimulate gametogenesis and reproductive behaviors [3]. The administration of exogenous hormones is regularly needed due to the lack of key environmental cues in captive settings that are necessary to stimulate reproductive events in amphibians. In addition to hormone therapy, ART protocols include, ultrasonography, in vitro fertilization (IVF) and sperm cryopreservation, all of which have been beneficial in the production of numerous threatened species [2,4,5].

One of the most promising technologies for managing threatened amphibian populations with declining heterozygosity is cryopreservation of sperm. The sperm cells held in cryo-storage can remain frozen for multiple years and can subsequently be thawed for use in generating future populations, as well as recovering lost genetic lineages. During freezing, sperm cells experience osmotic stress, ice nucleation, solution effects and cryoprotectant toxicity, which are all physical stressors that result in cell degradation and death [6]. Cells that survive the freeze–thaw cycle may exhibit molecular damage including disruption of disulfide bonds, DNA strand breaks, alteration of mRNA expression, changes in methylation patterns, and other epigenetic markers [7,8]. However, there are a number of antioxidants, membrane stabilizers, ice recrystallization inhibitors, antifreeze proteins and other compounds known to minimize ice nucleation and prevent physical and molecular stressors during cryopreservation, thereby mitigating deleterious effects of offspring from cryobanked sperm [6,7,8]. Examples of these mitigating compounds are permeating and non-permeating cryoprotectants, which have been shown to increase membrane fluidity and reduce water retention within the cell, minimizing ice crystal formation and its subsequent damage [9]. While various cryoprotectants, cooling rates, and thawing times are being investigated on post-thaw sperm survival [1], determining the reproductive potential of individuals produced using frozen sperm is critical as a proof of concept towards multigenerational sustainability. Further inclusion of ART into conservation breeding programs will depend on whether cryo-produced animals contain epigenetic defects that may reduce growth or reproductive potential in future generations. 

While we are unaware of any research looking at the epigenetic effects of amphibian offspring produced using frozen–thawed sperm, sperm cryopreservation in mammals has been shown to affect the expression of vital genes required for subsequent fertilization and offspring development [7]. These possible cryo-induced effects raise questions regarding the developmental competence of cryo-produced offspring and whether they are fit to produce offspring themselves. If cryo-produced offspring are not reproductively viable, then releasing individuals into the wild in support of recovery efforts is no longer a sustainable management solution, as they will not be able to add to the breeding pool of wild populations. While genetic analysis of offspring is incredibly important to help answer questions related to cryo-induced cell damage, determining the reproductive viability of cryo-produced offspring and their ability to produce a fit F2 generation is an essential part in determining the overall benefits of sperm cryopreservation. To date, most studies only report on the early development of amphibian embryos fertilized with cryopreserved sperm, in particular the rate of embryonic cleavage through hatching of swimming larvae [1,10,11,12,13,14]. There is only one study that measured reproductive viability through the production of an F2 generation; yet this study used testis macerates from euthanized males. The primary advantage we offer here is the collection of sperm from live male donors instead of testis macerates from deceased males [15]. The use of live males is a more sustainable solution as euthanasia is not feasible for threatened species whose genetics are extremely valuable. Furthermore, these animals may be part of established conservation breeding programs. The use of multiple techniques to retain and potentially increase genetic variation in these conservation programs is imperative for long-term population sustainability and management.

In this brief communication, we describe three amphibian species in which a suite of ARTs including exogenous hormone therapy, sperm cryopreservation, ultrasonography, and IVF were used to determine the fertilization potential of frozen–thawed sperm for the production of F1 offspring, followed by determining the reproductive viability of these cryo-produced offspring by testing their ability to produce an F2 generation. The three species tested in this study were the eastern tiger salamander *(Ambystoma tigrinum,* Least concern), the dusky gopher frog (*Lithobates sevosa*, Critically endangered) and the Puerto Rican crested toad (*Peltophryne lemur,* Endangered). The main objective of this report is to demonstrate that animals produced from frozen–thawed amphibian sperm, collected non-lethally, are reproductively viable, creating an F2 generation.

## 2. Materials and Methods

### 2.1. Animals

*A. tigrinum* and *L. sevosa* were maintained at the Conservation Physiology Lab at Mississippi State University. Animals were housed in same-sex tanks (30 cm × 46 cm × 66 cm) in groups of 3–6 per tank. Tanks were fitted with 3 inches of coco-fiber substrate, a water dish, and hides. In addition to ambient room lighting, type B ultraviolet light was provided for each tank and set on a 12 h light cycle to mimic the natural day light within their range in Mississippi (33.453880 latitude, −88.794090 longitude). Ambient temperature averaged 20.5 °C year-round. Animals were fed an alternating diet of crickets (*Gryllodes sigillatus*), mealworms (*Tenebrio molitor*), and Dubia roaches (*Blaptica dubia*) three times a week and were supplemented with calcium (Zoo Med Laboratories, Inc., San Luis Obispo, CA, USA) and a vitamin mix (Supervite, Repashy Ventures Inc., Oceanside, CA, USA) once a week by lightly dusting the insects immediately prior to feeding [16,17].

Male *P. lemur* were collected in June 2018 from Guayanilla, Puerto Rico. Once collected, males were held in soil-filled terrariums for no longer than 24 h. Prior to hormone administration, male toads were transferred to individual plastic tubs with 1–2 cm water for the duration of sperm collection to encourage urine production. Female *P. lemur* were housed at the Fort Worth Zoo (FWZ), in 15-gal tanks equipped with a mesh-wrapped eggcrate, raised false bottom, and a 25 × 15 cm hiding cover. Tanks were inclined to allow for a ~2 cm pool of reconstituted reverse osmosis (R-R/O) water, creating a 1:4 water to land ratio. Adult toads were kept on a 12-hour light cycle year-round, with a temperature gradient of 23.3–27.8 °C and basking areas at ~32.2 °C. Adults were fed gut-loaded and vitamin dusted crickets 3× weekly and a pinkie-mouse monthly [5].

All husbandry and experimental protocols for the *L. sevosa* and *A. tigrinum* were reviewed and approved by the IACUC at Mississippi State University (IACUC #13-109, #16-406, #17-189; #19-345; #20-160). *P. lemur* work was covered under IACUC #17-H001 at Fort Worth Zoo and animal collection was authorized through the FWZ-USFWS permit #TE121400-7.

### 2.2. F1 Animal Production: Sperm Collection and Cryopreservation

Males of each species were treated with exogenous hormones to elicit sperm production. Male *A. tigrinum* (*n* = 10) were intramuscularly administered a priming dose of 0.025 µg/g body weight (BW) gonadotropin-releasing hormone analog (GnRHa; Sigma-Aldrich, St. Louis, MO, USA, Product #L4513) and a spermiation dose of 0.1 µg/g BW GnRHa 24 h later, as previously described [17]. Spermic milt was expressed from male *A. tigrinum* by applying pressure to their lower abdomen, with collections starting at 1 h post hormone administration and continuing for 72 h [17]. Male *L. sevosa* (*n* = 6) were intraperitoneally administered a spermiation dose of 500 IU human chorionic gonadotropin (hCG; Sigma-Aldrich, St. Louis, MO, USA, Product #CG5) + 15 µg GnRHa [3]. Spermic urine and collected between 30 min and 2 h post hormone administration, as previously described [18]. Wild-caught male *P. lemur* (*n* = 10) were hormonally treated with either 10 IU/g BW hCG + 0.4 µg/g BW GnRHa via intraperitoneal administration (*n* = 7) or 10 µg GnRHa administered intranasally (*n* = 3). Spermic urine was collected at 1, 2, 3, 5 and 7 h post hormone administration, as previously described [19]. The volume of spermic milt or urine was recorded after each collection, and the samples were placed in sterile 1.5 mL Eppendorf tubes and stored at 4 °C. Immediately following sperm collection from each of the species, sperm motility and concentration were analyzed as described [1,3] with an Olympus CX43 phase contrast microscope (Olympus/Hunt Optics & Imaging INC., Pittsburg, PA, USA) to serve as a benchmark for post-thaw sperm assessments. Immediately following sperm collection, sperm motility was categorized as forward progressive motile (FPM, flagella-induced forward motion); non-progressive motile (NPM, undulating flagellum without forward propulsion); and nonmotile (NM). Total motility was calculated as the sum of FPM and NPM sperm. Concentration was determined using a hemocytometer (Hausser Scientific #3200, Horsham, PA, USA). For anurans, only spermic samples with a concentration of > 0.8 × 10^6^ sperm/mL were frozen, as this has been determined as an approximate threshold in sperm concentration needed for fertilization success [5,18]. For salamanders, sperm samples were brought to a concentration of 1 × 10^6^ sperm/mL before the cryopreservation process [17,20].

For each species of amphibian, variable sperm cryopreservation solutions and techniques have been tested and post-thaw sperm motility, fertilization capacity and survivability of the F1 generation reported previously [1,17,18]. Following pre-freeze analysis, all spermic samples were separated into 100 µL aliquots and mixed in a 1:1 ratio with species-specific cryoprotectant solutions. For *A. tigrinum*, sperm were combined with a cryoprotectant of 5% dimethyl sulfoxide (DMSO) + 0.5% bovine serum albumin (BSA) [17]. For *L. sevosa,* sperm were mixed with 10% N, N-dimethylformamide (DMFA) + 10% trehalose [18], and *P. lemur* sperm with 10% DMSO + 10% trehalose or 10% DMFA+ 10% trehalose [1,5]. Stock solutions were prepared at 2× the final concentration [5,18]. The resulting 200 µL cryosuspensions were equilibrated at 4 °C for 10 min before loading into 0.25 mL plastic freezing straws (Minitube International^®^, Tiefenbach, Germany) and plugging with Critoseal^®^ 223 (Leica BioSystems, Buffalo Grove, IL, USA). Straws were placed on a rack 10 cm above liquid nitrogen in a polystyrene freezing box, resulting in a freezing rate of −20 to −29 °C/min. After 10 min in liquid nitrogen vapor, straws were fully submerged in the liquid nitrogen and immediately added to the National Amphibian Genome Bank at Mississippi State University for long-term storage.

### 2.3. F1 Animal Production: Ultrasonography and In Vitro Fertilization

In preparation for the IVF attempts, ventral ultrasound scans of all female amphibians were carried out prior to hormone administration to assess natural follicular development for targeted hormone therapy [16,17]. A grading scale of 0–3 was utilized to classify egg development observed through sonography exams, with grades 0–1 representing little to no follicular development and grades 2–3 indicating that follicles are maturing and preparing for oviposition. In the early stages of folliculogenesis, immature oocytes are difficult to visualize and appear as small, white, grainy flecks through ultrasound imaging. As oocytes mature and undergo vitellogenesis follicles are separated by black area (egg jelly) and become very distinct and easy to visualize through imaging [21]. Ovarian development stages used to assess follicular maturity are depicted in Figure 1.

Female *A. tigrinum* (*n* = 5) were administered a hormone dose of 4 IU/g BW hCG + 0.1 µg/g BW GnRHa intramuscularly to induce oviposition. Eggs were manually expressed by applying pressure to the lower abdomen of the female, as previously described [17]. Approximately 10–20 eggs were placed into Petri dishes, and 20 µL of frozen–thawed sperm was applied per 10 eggs. Cryopreserved *A. tigrinum* sperm was thawed by submerging a straw in a 40 °C water bath until ice crystals melted (~5 s). Contents of the straw were expelled onto a Petri dish and immediately placed on eggs. A portion of the thawed sperm from the straws was kept to analyze post-thaw sperm motility, morphology, and viability. Results on the post-thaw viability of these samples have been published previously for *A. tigrinum* [17], *L. sevosa* [18] and *P. lemur* [5].

Female *L. sevosa* (*n* = 5) were intraperitoneally administered an ovulatory dose of 0.4 µg/g BW GnRHa + 13.5 IU/g hCG to induce oviposition and placed in separate containers with approximately 2 cm of water. Females were constantly monitored for the presence of eggs in the tubs. Egg-laying began 2–3 days post hormone administration, with eggs expressed by applying pressure to the abdomen of females, simulating a male in amplexus. Eggs (*n* = 20–30) were expressed into Petri dishes and fertilized with 100 µL of frozen–thawed sperm per dish. *P. lemur* females (*n* = 3) were intraperitoneally administered a hormone dose of 0.4 µg/g BW GnRHa to induce oviposition and placed in separate tubs with 2 cm of water. Females were monitored until eggs were seen in tubs. Egg-laying began approximately 10–12 h post-hormone administration, and eggs were expressed similar to *L. sevosa*. On average, 100 eggs were placed into separate Petri dishes, and approximately 100 µL of frozen–thawed sperm was used to fertilize the eggs. Anuran sperm was thawed similar to *A. tigrinum*; however, anuran sperm was diluted 1:10 with sterilized embryo transfer water (Sigma Aldrich, St. Louis MO, USA) to lower osmolality < 75 mOsmol/kg for sperm reactivation, prior to placing on eggs.

Fresh sperm from all three species was collected and applied to a subset of each species’ eggs to serve as a control for egg quality. The fresh sperm was collected from hormonally induced males following the same hormone treatments as described above. The percent of cleaved eggs, neurulas, and hatched tadpoles for each species are shown in Table 1 and represent the production of the F1 generation of animals.

### 2.4. F2 Production: Gamete Collection and In vitro Fertilization/Assisted Breeding

The life history of all three species used in this study indicate that on average, they reach sexually maturity within two years of age. Ultrasonography was utilized to assess the presence of egg follicles in females and morphological characteristics were used to determine sexual maturity in males (i.e., presence of thumb pads). Once the F1 offspring produced from cryopreserved sperm reached sexual maturity, individuals were chosen for assisted breeding to validate their reproductive viability. The selected F1 *A. tigrinum* female was serially evaluated using ultrasonography until she was determined to have mature egg follicles (i.e., grade 3, Figure 1). This female and two F0 male *A. tigrinum* were hormonally induced to stimulate gamete production, and an IVF was carried out as described above (Figure 2). Only fresh sperm from the two *A. tigrinum* males was used for this breeding event.

For the anurans, one F1 *L. sevosa* and one F1 *P. lemur* male were hormonally induced to elicit spermiation as described above. Ultrasonography was used to assess and select mature F0 females from both anuran species. To induce oviposition, hormones were administered to F0 females as described above. An IVF was used to test reproductive viability in the *L. sevosa*, and a facilitated breeding event was used for the *P. lemur* (Figure 2). For facilitated breeding, the selected F0 female and F1 male were placed in a breeding tank following hormone administration and left overnight to encourage natural reproductive behaviors (Figure 2). 

## 3. Results

Pre-freeze sperm motility was higher (*p* < 0.05, as reported in [5,17,18]) than post-thaw motility for *A. tigrinum* (50.4 ± 3.8% vs. 9.4 ± 2.1%), *L. sevosa* (90.0 ± 3.6% vs. 72.5 ± 2.5%), and *P. lemur* (67.0 ± 5.0% vs. 28.0 ± 3.0% for DMFA and 71.0 ± 5.0% vs. 25.0 ± 5.0% for DMSO), respectively. The number of offspring produced using fresh vs. frozen–thawed sperm was compared with determine the effect of cryopreservation on sperm fertilization potential, and standardize egg viability. In both *A. tigrinum* and *P. lemur,* hatch rates of offspring produced using frozen–thawed sperm was only 12% less than rates from fresh sperm. Interestingly, in *L. sevosa* the number of hatched tadpoles was actually 3% higher using cryopreserved sperm than fresh sperm (Table 1). The average cleavage rate of eggs fertilized with frozen sperm ranged from 4% to 48% depending on the species, compared with a range of 19–80% from fresh sperm. From these studies, we obtained 21 *A. tigrinum* larvae, 42 *L. sevosa* tadpoles and 46 *P. lemur* tadpoles from eggs fertilized with frozen–thawed sperm (Table 1).

Furthermore, the subset of the F1 offspring that was selected for breeding to determine the reproductive viability of cryo-produced individuals successfully produced offspring for every species. The IVF events designed to produce an F2 generation for *A. tigrinum* and *L. sevosa* resulted in the hatching of 5 larvae from 129 eggs and 48 tadpoles from 75 eggs, respectively. For the *P. lemur* facilitated breeding, 5095 tadpoles were produced (Table 1). We anecdotally observed that there can be substantial variation in fertilization rates depending on the quality of the eggs, as well as neurulation rates depending on water quality, or bacterial contamination in the egg jelly due to non-sterile environments. While this variation may be part of the reason for lower hatch rates, the main take away from the communication is to demonstrate the reproductive viability of F1 cryo-produced animals.

## 4. Discussion

Overall, this communication summarizes the success of incorporating ART, specifically hormone therapy, ultrasonography, sperm cryopreservation, and IVF into ex situ breeding programs for both caudate and anuran species. These results are particularly encouraging for widespread application of reproductive technologies for use in conservation breeding programs and long-term genetic management of declining amphibian populations. Moreover, this study represents one of the first reports of F2 generations of amphibians produced using cryopreserved sperm collected from live animals, for both anuran and caudate species. While all offspring for *A. tigrinum* and *L. sevosa* were maintained in their respective captive colonies, of the *P. lemur* tadpoles produced, tenwere kept within the captive breeding program for genetic management and, the remaining 5085 tadpoles were transported to Puerto Rico and released to the wild [5]. The *P. lemur* offspring produced from this study are genetically unique and may represent a new potential founder base in the captive colony, while also expanding the gene pool of the wild population. This addition of genetic diversity is incredibly important for the *P. lemur,* as there is a substantial risk for inbreeding to continue to increase in captive and wild populations [5].

The F2 tadpole production is quite high for both anuran species, with 64% and 98% of eggs developing into tadpoles in *L. sevosa* and *P. lemur,* respectively. This indicates that cryo-produced anuran males have functionally normal sperm that can effectively fertilize eggs and produce large numbers of offspring, similar to fresh sperm. While the offspring production rate yielded lower success for *A. tigrinum* (4%) than for the two anurans, the fact that individuals were produced shows that the female generated from frozen–thawed spermatozoa was reproductively viable and marks another conservation milestone in amphibian ART (Table 1). The large drop-off seen in survival between the neurulation rate and hatching rate does not seem to be a unique characteristic to embryos fertilized with cryopreserved sperm. A similar decline was observed for the fresh sperm controls, indicating that egg quality, or lack of sterile environment associated with embryo development may be involved. 

While an F1 female salamander produced from cryopreserved sperm was used to show reproductive viability through egg production following IVF, F1 males were utilized for the two anuran species. Thus, both male and female F1 amphibians were able to produce an F2 generation using ART. In a separate study, our lab has shown that two cryo-produced female Houston toads (*Anaxyrus houstonensis*) were able to produce an F2 generation, indicating that within the anuran group, both males and females generated with frozen–thawed sperm have now produced offspring [21]. This report of sexually mature adults derived from cryopreserved sperm that can produce their own offspring, represents an important proof of concept where we demonstrate the viability of ART as a conservation tool for genetic management. In addition, more IVF trials using frozen–thawed sperm, conducted in parallel with fresh sperm, can better characterize whether any developmental differences are observed due to the impacts of freezing on sperm integrity. 

Our results are further supported by other studies that have not found differences in breeding success between cryo-produced and control adults. For example, Pearl et al. (2017) did not find any variation in reproductive success between *Xenopus laevis* derived from frozen–thawed sperm and control animals. Furthermore, *X. laevis* derived from frozen–thawed sperm produced 90% phenotypically normal sperm, displayed typical patterns of gene expression, grew at the same rate, and had similar mass at adulthood [15]. Overall, the results described in these studies provide encouraging insight on the success of ART in captive breeding programs, specifically in producing reproductively viable offspring.

## 5. Conclusions

Determining the reproductive viability of cryo-produced individuals is a critical aspect of species recovery programs, specifically those that include reintroduction as a mitigation effort. Reintroducing non-reproductive individuals to the wild is counter-intuitive to the overall goals of these programs, as these individuals would be unable to breed in the wild and would therefore not assist in population and genetic management of the population. In this study, we showed that cryo-produced individuals are in fact reproductively viable, indicating that those released to the wild may be able to assist recovery. The next step in determining the success potential of these reintroduced animals is establishing if there are any lingering effects of cryopreservation on future generations, such as reduced size, delayed reproductive maturity, and lower competition levels. Overall, the advancements in amphibian sperm cryobiology described here, for both anurans and caudate species, reinforce the importance of continuing innovation and application of cryobiology, and genome resource banks as a conservation tool to help mitigate the rapid loss of amphibian biodiversity.

## Figures and Tables

**Figure 1 animals-13-00053-f001:**
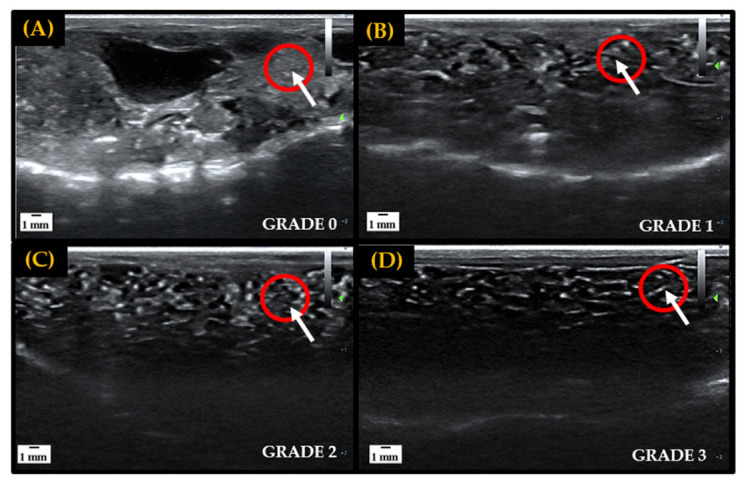
Representative ultrasound grading scale from 0 to 3 utilized to track and determine follicular maturity for breeding in female amphibians. Fluid, or egg jelly, is indicated by anechoic area (black), and hyperechoic areas (white) represent egg follicles, tissue, bones or other denser components. The red circles indicate the gradual maturation of follicles in a localized area. Panel (**A**): Large hyperechoic area indicates the absence of egg jelly surrounding follicles, signifying a lack of development—Grade 0. Panel (**B**): Increase in anechoic area, and slight separation of follicles, indicates the accumulation of egg jelly as follicles start to develop—Grade 1. Panel (**C**): Hyperechoic area is spread out and anechoic space begins to further separate follicles, indicating that more egg jelly is being recruited and follicles are developing—Grade 2. Panel (**D**): High density of anechoic area separating follicles indicates matured egg follicle—Grade 3.

**Figure 2 animals-13-00053-f002:**
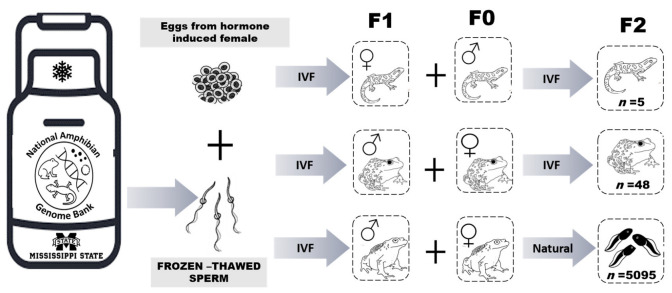
Schematic diagram of the Assisted Reproductive Technologies process used to create the F1 generation of adults and subsequent production of F2 progeny. IVF = in vitro fertilization and Natural = animals stimulated with hormones for natural breeding. F0 indicates animals not produced using cryopreserved sperm, F1 is the first generation produced using cryopreserved sperm, and F2 is the generation representing offspring from breeding the F1 and F0 animals.

**Table 1 animals-13-00053-t001:** Summary table of cleaved embryo and larvae/tadpole development totals for the F1 and F2 generations produced through cryopreservation and in vitro fertilization. Eggs numbers for each species represent one male and female pairing. Cleaved embryos are shown as a percentage of eggs used, whereas neurula and hatch rates are shown as a percentage of cleaved embryos. Missing values for *A. tigrinum* and *P. lemur* cleavage rates were due to challenges associated with access to zoo or academic buildings overnight.

Embryo Development Totals				
		F1 Control IVF	F1 Production	F2 Production
Species	Development stage	Fresh Sperm	Cryopreserved sperm	Gametes from F1 adults
		No. (%)	No. (%)	No. (%)
*Ambystoma tigrinum*				
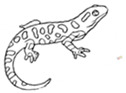	Eggs	267	256	129
	Cleaved	126 (47%)	64 (25%)	-
	Neurula	85 (67%)	37 (58%)	10
	Hatched	57 (45%)	21 (33%)	5

*Lithobates sevosa*				
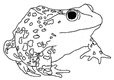	Eggs	988	1341	75
	Cleaved	792 (80%)	605 (48%)	58 (77%)
	Neurula	502 (63%)	290 (48%)	48 (83%)
	Hatched	33 (4%)	42 (7%)	48 (83%)

*Peltophryne lemur*				
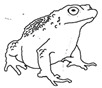	Eggs	2691	6981	5176
	Cleaved	525 (19%)	306 (4%)	-
	Neurula	215 (40%)	55 (18%)	-
	Hatched	140 (27%)	46 (15%)	5095

## Data Availability

Data are available upon request from Carrie K. Kouba (ckv7@msstate.edu).

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
