# Peer review of "Sperm Cryopreservation as a Tool for Amphibian Conservation: Production of F2 Generation Offspring from Cryo-Produced F1 Progeny"

_animals, 2022, doi:10.3390/ani13010053_

Round 1
Reviewer 1 Report
This manuscript reports the use of frozen-thawed sperm for artificial breeding of 3 amphibian species. Cryopreservation optimized for amphibian species are crucial steps for developing protection strategies for these endangered animals.
Overall, the manuscript is well written. The experimental methodology is sound, and the data are carefully and rigorously analyzed.
One minor comment concerns the cryoprotectants that are used for obtaining efficient sperm freezing and thawing is critical. It would be good to provide a bit more information about the different mixtures optimized and used for each species. What was the rational and if possible, how their effectiveness was evaluated. It would be useful to mention other mixture that did not work or were less efficient. Perhaps, this information could be listed in a table.
Other minor points:
1) Abstract Line 23: describes (rather than highlights). It may look as if the authors did not do any works but report work of others. Also suggest “generated” instead of “created”.
2) Figure 1: please add scale bars on the pictures
Reviewer 2 Report
Abstract
- The abstract lacks a clear and well-structured study purpose, instead it only describes the results obtained from 3 types of amphibians. Is this due to the type of report (Brief Report)?
Introduction
- The introduction is brief and informative and highlights the importance of the study. I would only suggest that the authors mention at the end of the introduction the main objective of the study rather than the hypothesis.
Materials and methods
- Lineas 135 – 137: The authors describe a method for semen collection, but it is not referenced. is the method standardized and validated by the authors?
- Linea 139: Why do the authors refer to a sperm urine? Is it not possible to obtain semen in this species?
- Linea 146 – 147: The authors describe that immediately after sample collection they assessed motility and concentration, but do not describe protocols and/or equipment used (microscope, CASA, automated counter, cell counting chamber).
- Linea 148: At what concentration did the authors adjust the samples before freezing?
- Lineas 149 – 153: The authors describe that aliquots were mixed in a 1:1 ratio with species-specific cryoprotectant solutions. But they do not describe in detail these solutions or the commercial name they have. The authors only indicate how they were supplemented with DMSO and BSA. I recommend the authors to make a more detailed description of the cryopreservation and analysis procedure as this technology is a central topic in their study.
Línea 158: only for a correct interpretation of the procedure I recommend the authors to mention "nitrogen vapours".
Results
- It would be useful for the manuscript if the authors could add a table with values for motility, morphology and viability of fresh semen before freezing and values for these same variables post-thawing, regardless of whether they state that they have been published previously.
- Lineas 251 – 257: I recommend that the authors only describe the results in this section. Any discussion of them should be done in the respective section.
Conclusions
- Conclusions are statements that based on the results and discussion allow us to establish something as true, valid, or possible. Therefore, I recommend that authors considerer edit this section.
